# Antibacterial Potential of *Allium ursinum* Extract Prepared by the Green Extraction Method

**DOI:** 10.3390/microorganisms10071358

**Published:** 2022-07-06

**Authors:** Alena Stupar, Ljubiša Šarić, Senka Vidović, Aleksandra Bajić, Violeta Kolarov, Bojana Šarić

**Affiliations:** 1Institute of Food Technology, University of Novi Sad, Bulevar Cara Lazara 1, 21000 Novi Sad, Serbia; alena.stupar@fins.uns.ac.rs (A.S.); aleksandra.bajic@fins.uns.ac.rs (A.B.); bojana.saric@fins.uns.ac.rs (B.Š.); 2Faculty of Technology, University of Novi Sad, Bulevar Cara Lazara 1, 21000 Novi Sad, Serbia; senka.curcin@yahoo.com; 3Faculty of Medicine, University of Novi Sad, Hajduk Veljkova 3, 21000 Novi Sad, Serbia; violeta.kolarov@mf.uns.ac.rs

**Keywords:** *Allium ursinum*, antimicrobial activity, subcritical water extraction, sulfur compounds, polyphenolics

## Abstract

The antimicrobial activity of *Allium ursinum* aqueous extract prepared using high pressure extraction was evaluated. Minimal inhibitory concentrations (MIC) and minimal bactericidal concentrations (MBC) of *A. ursinum* extract for six bacterial pathogens were determined using the broth macrodilution method. Although the *A. ursinum* extract was shown to be effective toward all investigated foodborne bacteria, its antimicrobial activity depended on its concentration and bacterial strain. *Listeria monocytogenes* was the most sensitive to antimicrobial activity of *A. ursinum* extract among all tested pathogens. Accordingly, the lowest MIC and MBC of *A. ursinum* extract were determined for *L*. *monocytogenes* (28 and 29 mg/mL). The tested extract showed a similar antimicrobial potential to other examined bacterial strains (*Salmonella* Enteritidis, *Proteus hauseri*, *Enterococcus faecalis* and two strains of *Escherichia coli*) with MIC and MBC values at concentrations of 29 and 30 mg/mL, respectively. The dependence of the antimicrobial activity of the *A. ursinum* extract on the level of contamination of tested pathogens was also observed. The increase in the contamination level caused an intense reduction in antibacterial potential of the *A. ursinum* extract. The composition of the *A. ursinum* extract was analyzed and found to be a good source of polyphenols and sulfur compounds. However, considering the applied extraction method and the HPLC analysis of bioactive compounds, the antimicrobial potential may be attributed more to polyphenol content. The obtained results that the extracts have shown toward food pathogens open the possibility of using the tested extracts as natural additives in a variety of food products.

## 1. Introduction

Increasing awareness about the benefits of a healthy diet has led to increased demands for food without chemical preservatives. The growing demand for food that does not contain synthetic additives is fully justified, addressing the increased resistance of microorganisms to commercial antimicrobial agents, but also regarding the consumer concerns about the safety of used synthetic additives [1,2]. For this reason, extensive research has been made to find natural antimicrobial agents and additives in order to improve the quality and shelf-life of the food products. Researchers have increasingly focused on studying natural products and compounds that could replace synthetic ingredients in food systems. Plant extracts have attracted considerable attention owing to their bioactive compounds that may have various functional properties, including antimicrobial activity. Application of such plant extracts in food products can improve the nutritional profile of foods and can ensure food safety as well as enable the creation of novel functional products and dietetic supplements [3].

*Allium* species have already been recognized due to their antimicrobial activity against bacteria and fungi; however, most research is focused on the antimicrobial activity of garlic and onion as representatives of this species [4,5]. In the past few years, scientific interest for *Allium ursinum*, known as wild garlic or bear’s garlic, has significantly grown. *A. ursinum* has been used in traditional medicine for centuries. Several health benefits have been associated with *A. ursinum*: cardioprotective, antiplatelet, antidiabetic, antioxidative, antimicrobial, and anti-inflammatory [6,7]. These benefits have been attributed mainly to sulfur-containing compounds such as thiosulfinates and (poly)sulfides; however, the health benefits and strong antioxidant activity can also be attributed to the polyphenolic content [8,9]. Sulfur-containing compounds of wild garlic are mainly responsible for its traditional use in terms of culinary and medicinal purposes. In folk medicine, *A. ursinum* is mostly consumed as an aqueous or ethanolic extract prepared with domestic spirits by maceration [10]. To scientifically confirm the effects of traditional use and folk medicine, it is of the interest to use the same solvents with at least similar conditions of preparation. However, the conventional method of extraction mostly results in a low yield of bioactive compounds while using large amounts of solvents and plant material, therefore generating a large amount of waste. To isolate the molecules of interest and to assure their utilization as a food additive, it is necessary to apply a suitable extraction method that will preserve and increase the yield of the targeted compounds. Principles based on green chemistry have been gaining widespread attention not only on a scientific level, but also on the industrial and consumer level. Therefore, it is important to bring more sustainable approaches into the extraction of bioactive compounds and natural food additives through ecofriendly strategies and to fulfil the increasing consumer request for greener products, cleaner labels, and sustainable processes.

Herein, we report an efficient procedure for the extraction of bioactive compounds from *A. ursinum*, addressing the principles of green processing using water as a solvent at its subcritical condition. Subcritical water could be an excellent alternative medium for the extraction of target compounds in food and herbal plants. Using water as a safe, green solvent at its supercritical condition, the obtained extracts are higher quality and have bioactivity [11,12]. The extracts thus obtained can be directly incorporated into the final products; that is, there is no need for their purification, which makes the process more economical.

The objective of this study was to examine the antimicrobial activity of *A. ursinum* extract prepared by subcritical water extraction toward selected foodborne pathogens. Although *A. ursinum* extracts obtained by various solvents have already been discussed [12,13,14], according to our knowledge, there are no available literature sources on the antimicrobial properties of *A. ursinum* extracts prepared by subcritical water extraction. Accordingly, the effects of the level of contamination on antimicrobial capacity of *A. ursinum* extract were examined. Since the antibacterial activity of *Allium* species is generally ascribed to sulfur and polyphenolic compounds, total phenolic and flavonoid content was determined. The composition of phenolic and sulfur compounds in *A. ursinum* extract was determined by HPLC.

## 2. Materials and Methods

### 2.1. Chemicals

Folin–Ciocalteu reagent was purchased from Sigma (Sigma-Aldrich GmbH, Sternheim, Germany). Used chromatographic standards were as follows: allicin, AC (ChromaDex, Inc., Los Angeles, CA, USA), allyl sulfide, AS (Sigma-Aldrich GmbH, Germany), diallyl sulfide, DADS (Sigma-Aldrich GmbH, Germany) and methanethiosulfonic acid S-methyl ester (MMTS2). Phenolic standards such as gallic acid, caffeic acid, ferulic acid, chlorogenic acid, *p*-coumaric acid, kaempferol and quercetin were purchased from Sigma-Aldrich (Steinheim, Germany). Catechin, epicatechin, epigallocatechin gallate, epicatechin gallate were purchased from Extrasynthese (Genay, France). All other chemicals and reagents were of analytical and HPLC reagent grade, while ultrapure water was obtained from a Milli-Q system (Millipore, Billerica, MA, USA).

### 2.2. Extract Preparation

Wild garlic leaves were collected at Fruska Gora mountain, Serbia. Collected leaves were sorted, washed in water, were frozen at −20 °C, freeze-dried, sealed in bags, and stored in the dark at room temperature in a desiccator. Prior extraction plant material was ground in a kitchen blender (0.325 mm), and moisture content (6.12%) was determined. Wild garlic extract was prepared following the recommended optimal conditions investigated by Tomšik et al. (2017) with slight modification [13]. The extraction was performed in ASE 350 system Dionex Corporation (Sunnyvale, CA, USA). Powdered wild garlic samples were placed into the extraction cell together with diatomic earth, and extraction was performed with water as solvent at 180 °C/1500 psi for 10 min. The extract was stored at 4 °C until further analysis.

### 2.3. Extract Characterization

#### 2.3.1. Determination of Total Phenolic Content (TPC) and Total Flavonoid Content (TFC)

Total phenolic content was determined using a spectrophotometric method based on the color reaction of phenols with Folin–Ciocalteu’s reagent, according to the procedure proposed by Singleton and Rossi (1965) [15]. Total flavonoid content was also detected using spectrophotometric method based on the color reaction of flavonoids with aluminum chloride [16]. The total phenolic content results were estimated as gallic acid equivalents per gram of dried weight (mg GAE/g DW), and total flavonoid content was expressed as mg quercetin equivalents per gram of dried weight (mg QE/g DW). Analyses were performed in three replicates.

#### 2.3.2. Determination of Phenolic Compounds by HPLC

Chromatographic analyses were performed using Agilent 1200 Liquid Chromatograph (Agilent, Paolo Alto, CA, USA), Agilent 1220 diode array detector (DAD) and Agilent Eclipse XDB-C18 column (4.6 × 50 mm, 1.8 μm). Chromatographic analyses were carried out under linear gradient with solvent A (methanol) to solvent B (1% formic acid in water) as follows: initial 85% B; 0–6.2 min, 85% B; 6.2–8 min, 85–75% B; 8–13 min, 75–61% B; 13–15 min, 61% B; 15–20 min, 61–40% B; 20–25 min, 40–0% B. A flow rate of 1 mL/min was set, while the run time and post-run time were set as 25 and 10 min, respectively. Prior analysis of A. ursinum extracts was diluted with solvent mixture of methanol and 1% formic acid in water (50:50, *v*/*v*) sonicated for 10 min. Solution was filtered using 0.45 μm regenerated cellulose membrane filters (Agilent, Paolo Alto, CA, USA), and 5 μL of extract was injected into the system. The spectra were recorded in the 190–400 nm range with chromatograms plotted at 280, 330 and 350 nm. Identification of phenolic compounds in the sample was based on a comparison of their retention times and spectral data with those of the standards. When standard was not available, the content of detected compound was expressed as corresponding phenolic compound equivalent [13].

#### 2.3.3. Determination of Sulfur Compounds by HPLC

Sulfur compounds were detected according to the method described by Tomšik et al. (2018) [8]. Liquid chromatograph (Agilent 1100 series, Paolo Alto, CA, USA), using CORTECS C18 column (internal diameter 100 × 100.6 mm, charge size 2.7 mm), with Waters VanGuard pre-column (internal diameter 5 × 3.9 mm; charge size 2.7 mm) and diode array detector (Agilent, USA) in stationary mode were used to determine sulfur compounds. The solvent flow was adjusted to 1 mL/min, and the injection volume was 50 µL. A mixture of phosphate buffer (20 mM, pH 4.5), 95% and acetonitrile 5% was used as the mobile phase. Chromatograms were recorded at a wavelength of 210 nm. Quantification was determined by integrating the obtained peaks and comparing the results with a series of standards of known concentrations by creating calibration curves.

### 2.4. Antimicrobial Activity

#### 2.4.1. Bacterial Strains and Growth Conditions

The antibacterial activity of the *A. ursinum* extract was tested against selected Gram-negative (*Escherichia coli* ATCC 10536, *Escherichia coli* ATCC 8739, *Salmonella* Enteritidis ATCC 13076, *Proteus hauseri* ATCC 13315) and Gram-positive bacteria (*Listeria monocytogenes* ATCC 19111, *Enterococcus faecalis* ATCC 29212). Lyophilized bacteria were stored in the refrigerator until the moment of activation. Reconstitution was performed according to the manufacturer’s instructions. Refrigerated slant cultures (Nutrient agar, Himedia, India) were sub-cultured weekly. Prior to each antibacterial test, selected bacterial strains were sub-cultured on nutrient agar (Himedia, India) and incubated at 37 ± 1 °C for 18–24 h. After incubation, bacteria were aseptically transferred to 0.1% peptone salt solution (Himedia, India) and well homogenized. The density of the bacterial suspensions was adjusted to the turbidity of a 0.5 McFarland standard using the DEN-1 densitometer (Biosan, Latvia). These initial suspensions were used to prepare further decimal dilutions in 0.1% peptone salt solution intended for preparation of artificially contaminated samples of *A. ursinum* extract.

#### 2.4.2. Antimicrobial Assay

The antimicrobial assay was conducted using the broth macrodilution method [17]. The *A. ursinum* extract was added to 1 mL of growth medium (Nutrient broth, Himedia, India) to give concentrations of 5, 10, 15, 20, 25 and 30 mg/mL. In order to obtain artificially contaminated samples at levels of 10^2^, 10^3^, 10^4^, 10^5^ and 10^6^ cfu/mL, the appropriate volume of inoculum from the selected decimal dilution was aseptically transferred into 1 mL of growth media with a defined concentration of *A. ursinum* extract. Each sample was separately contaminated with individual test bacteria. Artificially contaminated samples of subcritical water extract of *A. ursinum* in nutrient broth (Himedia, India) were incubated for 24 h at 37 ± 1 °C. Bacterial growth was determined by taking samples after incubation and plating on cultivation media (Plate count agar, Himedia, India) according to the standard method ISO 4833-1:2013. Artificially contaminated nutrient broths (Himedia, India) without added plant extract were used as positive controls, while non-inoculated nutrient broths (Himedia, India) containing extract of *A. ursinum* in selected concentrations were used as a negative controls. All experiments were conducted in triplicate.

#### 2.4.3. Minimal Inhibitory Concentration (MIC) and Minimal Bactericidal Concentration (MBC) Determination

After obtaining the results of the previously described antimicrobial assay of *A. ursinum* extract (5, 10, 15, 20, 25 and 30 mg/mL), the same extract at concentrations of 26, 27, 28 and 29 mg/mL was tested in order to determine the MIC and MBC. The lowest concentration of plant extract causing a reduction of more than 90% in inoculum viability after 24 h of incubation was reported as the MIC, while the MBC was defined as the lowest concentration of the plant extract that completely eliminates the (100%) tested bacteria [18]. The MIC and MBC determination were conducted as described in Section 2.4.2. All experiments were performed in triplicate.

### 2.5. Statistical Analysis

Results were expressed as mean ± standard deviations of triplicate analyses for all measurements. Analysis of variance was followed by Tukey’s post hoc test using STATISTICA version 10, Minitab 17 (StatSoft Inc., Tulsa, OK, USA). *p* values < 0.05 were regarded as significant.

## 3. Results

### 3.1. Chemical Characterization of A. ursinum Extract

The understanding of chemical composition and potential biological properties of plant extracts is crucial for the understanding of their biological activity and their further incorporation into the food matrix.

Total phenolic content in the examined extract was 4.23 mg GAE/100 g DW, while total flavonoids content was measured at 0.73 mg CE/100 g DW. Total phenolic and total flavonoids of the examined *A. ursinum* extract obtained by subcritical water extraction, as expected, showed to be several folds higher comparing to the extract obtained by conventional extraction, maceration. The total phenolic content showed to be more than 3.5-fold higher as the content of total phenols in the extract obtained by maceration (1.20 mg GAE/100 g DW), while the content of total flavonoids was about three-fold higher than flavonoids content obtained by maceration (0.22 mg CE/100 g DW).

Phenolic profile and qualitative analysis of *A. ursinum* extract was detected based on the available standards obtained from the spectra of phenolic acids and flavonoids, peak retention times and available literature data (Table 1). Based on the spectrum of dominant components in the tested extract, it can be assumed that the *A. ursinum* extract obtained by subcritical water extraction contains compounds such as phenylpropanoids and flavonoids. According to previous literature reports, the most abundant compounds in *A. ursinum* are various derivates of kaempferol [19]. Several kaempferol derivates with the concentration range of 1.97–89.19 μg/mL (expressed as kaempferol equivalent/mL extract) were also the dominant compounds in our investigated extract obtained by subcritical water. Their UV spectra showed characteristics of a flavanol-type structure with two absorption maxima nm similar to that of the standard, which indicates that those compounds are kaempferol derivatives (λmax 348). Additionally, the presence of catechins derivates (1.85–7.24 μg equivalent/mL extract) and gallic acid and its derivates (2.13–32.86 μg equivalent/mL extract) was confirmed. The main phenolic compound detected in the extract was the gallic acid derivate with a concentration of 32.86 μg GA equivalent/mL extract; however, flavonoids presented the majority of the total phenolic content detected by HPLC. Quantitative phytochemical analysis of major sulfur compounds revealed the presence of several dominant compounds. The most abound sulfur compounds in the subcritical water extract of *A. ursinum* was S-methyl methanethiosulfonate (302.6 μg/mL extract), followed by alilsulfid (44.1 μg/mL extract) and diallyl disulfide (27.3 μg/mL extract). These and other related compounds were already detected in *A. ursinum* extract [6]; however, as a totally different extraction was applied in the research, the compounds and their content cannot be compared.

### 3.2. Antimicrobial Activity of A. ursinum Extract

The results of the antimicrobial assay summarized in Table 2 clearly indicate the inhibitory effect of the *A. ursinum* extract on tested pathogens dependent on its concentration and bacterial strain.

The *A. ursinum* extract did not show antimicrobial potential at the lowest tested concentration (5 mg/mL). The strongest antimicrobial activity of this extract was observed at the concentration of 30 mg/mL since the tested bacteria were not detected after 24 h of incubation (Table 2).

*L. monocytogenes* was the most sensitive to antimicrobial activity of *A. ursinum* extract among all tested pathogens. The broth samples containing *A. ursinum* extract at a concentration of 10 mg/mL had a lower number of viable cells of this bacterium at the end of incubation compared with the positive controls (Table 2). It is obvious that the antimicrobial activity of *A. ursinum* extract (10 mg/mL) toward *L. monocytogenes* could be described as growth inhibitory since the extension of the *L. monocytogenes* lag phase and growth slowing were observed. It was particularly evident for contamination levels of 2–5 log_10_ cfu/mL where the count of *L. monocytogenes* after 24 h of incubation remained similar to the initial contamination. The highest count of *L. monocytogenes* at the end of incubation determined at the highest level of contamination (6 log_10_ cfu/mL) indicates that the antibacterial activity of the extract is also dependent on the level of contamination (Table 2).

Although the highest applied concentration of the *A. ursinum* extract (30 mg/mL) showed a bactericidal effect against all tested pathogens, this extract in lower concentrations (10, 15 and 25 mg/mL) possessed weaker antimicrobial potential toward *S.* Enteritidis and both strains of *E. coli*, *P. hauseri* and *E. faecalis*, in comparison to that exhibited against *L. monocytogenes*. However, when comparing the counts of these pathogens in broth samples containing *A. ursinum* extract at a concentration of 10 mg/mL to the counts of these bacteria in positive controls, it is clear that the extract of *A. ursinum* in this concentration still has a certain antimicrobial effect (Table 2). Namely, the number of tested pathogens in positive controls was statistically significantly higher (*p* < 0.05) than in broths with added extract, and the antimicrobial activity of the extract was reflected in a certain slowing down of the growth of bacteria. Both strains of *E. coli* tested in this study exhibited similar sensitivity to all applied concentrations of *A. ursinum* extract. Despite the fact that the slightly lower number of *E. coli* ATCC 8739 was detected in the tested samples, it cannot be clearly concluded that this strain is more susceptible to the antimicrobial activity of *A. ursinum* extract than *E. coli* ATCC 10536 since the number of *E. coli* ATCC 8739 was also slightly lower in the positive controls in comparison to *E. coli* ATCC 10536.

Results of the antimicrobial activity of *A. ursinum* extracts at concentrations of 15, 20 and 25 mg/mL were not shown because there was no statistically significant difference between their antimicrobial activity and the antimicrobial activity of the *A. ursinum* extract at a concentration of 10 mg/mL. The presence of tested bacteria was not detected in the negative controls.

### 3.3. Minimal Inhibitory Concentration (MIC) and Minimal Bactericidal Concentration (MBC) of A. ursinum Extract

The obtained results of the antimicrobial assay (Table 2) indicated that the MIC values of *A. ursinum* extract were within the range of concentrations of 25–30 mg/mL. Therefore, an additional antimicrobial test was performed, and the results are shown in Table 3.

*L. monocytogenes* was confirmed to be the most sensitive to the antimicrobial activity of the *A. ursinum* extract (Table 3). The lowest MIC of the *A. ursinum* extract toward tested pathogens was determined for *L. monocytogenes* (28 mg/mL). The tested extract showed a similar antimicrobial potential to other examined bacterial strains with MIC values at a concentration of 29 mg/mL (Table 3). The dependence of the antimicrobial activity of the *A. ursinum* extract on the level of contamination of tested pathogens was observed (Table 2). At the contamination levels ranging from 2 to 5 log cfu/mL, *S*. Enteritidis, both strains of *E. coli*, *P. hauseri* as well as *E. faecalis* were below the limit of quantification (<1 cfu/mL) after 24 h of incubation in samples containing *A. ursinum* extract at a concentration of 29 mg/mL. Conversely, in the experiments with the highest level of contamination (6 log cfu/mL), these pathogens at the end of incubation were still present in the samples with the same concentration of extract (29 mg/mL). This indicates a clear correlation between the intensity of antimicrobial activity of the *A. ursinum* extract and the level of contamination of the tested bacteria. The minimum bactericidal concentration (MBC) of *A. ursinum* extract against *L. monocytogenes* was 27 mg/mL for levels of contamination ranging from 2 to 4 log cfu/mL and 29 mg/mL for the levels of contamination of 5 and 6 cfu/mL (Table 3). The previously determined MBC of tested extract toward *S.* Enteritidis, *E. coli*, *P. hauseri* and *E. faecalis* (30 mg/mL) (Table 2) was confirmed in the second antibacterial test (Table 3) since these pathogens were still viable (at the highest level of contamination) at the end of the incubation at all tested extract concentrations (26–29 mg/mL). The presence of the tested bacteria was not detected in the negative controls.

## 4. Discussion

Foodborne illnesses are a major concern for consumers, the food industry, and food safety authorities [1]. The determination of chemical composition and potential biological properties of plant extracts is crucial for the understanding of their properties and their further use. Prior to the application of herbal extracts with potential antimicrobial effects in the food industry, it is necessary to test their antimicrobial activity.

It was considered that the chemical composition of *A. ursinum* is complex. The presence of about 100 compounds has been detected by various authors with different extraction methods and analytical methods of detection [6,14,20,21,22]. By comparing the results reported by different authors, it can be concluded that the chosen extraction technique and solvent significantly affect the profile of isolated compounds. Compared with the maceration, subcritical water extraction gave several-fold higher results of TPC and TFC, justifying its use for obtaining the extract with high bioactive content. Additional differences may occur depending on the plant origin, vegetation period and used herbal parts. The antimicrobial activity of the *Allium* species is mainly attributed to various kinds of alk(en)yl alka/ene thiosulfinates and their transformation products [6,23] and polyphenolic compounds. Even the fresh *A. ursinum* is characterized by the presence of its characteristic odor from sulfur compounds such as allicin. In our extract, obtained by subcritical water extraction, allicin was not isolated. The absence of allicin in the extract obtained by subcritical water extraction is most likely due to the thermolability of the sulfur compounds, given the high temperature applied in the extraction conditions and high allicin instability. Sulfur compounds such as S-methyl methanethiosulfonate, alilsulfid and diallyl disulfide detected in our extract arise as products of degradation when diallyl thiosulfinate converts to various sulfides, with diallyl disulfide (DADS) being the most abundant, while S-methyl methanethiosulfonate presents thermal breakdown products that contribute to the typical flavor of processed vegetables [24]. The number of available sulfur atoms is important in conferring the potency of antimicrobial activity. Thiosulfinates inhibit microorganisms because of their –S(O)–S– group, which generally reacts with the SH group of cellular proteins to generate mixed disulfides [25]. In addition, lipid synthesis is affected, and the phospholipid biolayer of the cell wall cannot form correctly in both Gram-positive and Gram-negative bacteria [25].

In addition to the sulfur compounds, biomolecules such as peptides, flavonoids, phenols, alkaloids, and saponins contribute to the antimicrobial activity of *Allium* species or reveal synergistic effects with other present bioactive compounds [7,26]. Polyphenolic compounds have great structural diversity and variations in chemical composition and thus differ in their antibacterial effectiveness against pathogenic microorganisms [1]. The mechanisms of antibacterial action of phenolic compounds are not yet fully deciphered, but phenolic compounds are known to involve many sites of action at the cellular level. For example, it is known that gallic acid has a strong antibacterial effect. It can induce irreversible changes in the membrane properties of *E.coli*, *Pseudomonas aeruginosa*, *S. aureus* and *Listeria monocytogenes* [27]. The antimicrobial activity mainly depends on the position of the hydroxyl and carboxyl groups and the double bonds present in the ring. The high antimicrobial activity of phenolic compounds also depends on the size of the added alkyl or alkenyl group [28]. Kaempferol and catechin antimicrobial activities were confirmed in several studies [29,30] This finding is important, as in our extract, the presence of gallic acid derivate, catechin and kaempferol derivates was confirmed, which contributed to the antimicrobial activity of the *A. ursinum* extract. According to the literature data, we may hypothesize that the presence of flavonoid constituents mostly contributes to the antioxidative potential of *A. ursinum* extract. Numerous studies support the fact that *A. ursinum* can be used as a natural antimicrobial agent. However, previous studies have shown conflicting results regarding the antimicrobial activity of *A. ursinum* extracts against various Gram-positive and Gram-negative bacteria [7,8,31,32,33]. As it was stated before, this can be explained by the isolation of different active compounds using different solvents during extraction, extraction method, plant origin and plant part. Accordingly, Ivanova et al. [33] tested the antibacterial activity of acetone, chloroform, ethyl acetate, n-butanol and water extracts of fresh flowers and leaves of *A. ursinum* against various Gram-positive and Gram-negative bacteria. None of the extracts tested showed antibacterial activity against Gram-negative *E. coli*, while acetone and chloroform extracts containing organosulfur compounds exhibited significant inhibition of Gram-positive *S. aureus*. Therefore, the authors who investigated antibacterial activity of *A. ursinum* extracts reported different antimicrobial activity depending on the extraction method, solvent and isolated bioactive compounds in the extracts [25,34]. Furthermore, the antibacterial activity of the tested material also depends on bacterial strain. Sapunjieva et al. [35] reported stronger antibacterial effects of 70% ethanol extract of *A. ursinum* on Gram-positive bacteria (*L. monocytogenes*, *S. aureus*) in comparison to Gram-negative bacteria (*E. coli*, *Salmonella* enterica subsp. Enterica serovar Abony) [35]. Synowiec et al. [36] investigated the antibacterial activity of water and methanol extracts of *A. ursinum* (at the concentration range 0.16–83.7 and 0.06–35.5 mg/mL, respectively) toward *S. aureus*, *B. subtilis*, *E. coli*, *P. mirabilis* and *S.* Enteritidis. The water extract of *A. ursinum* exhibited antimicrobial activity only against *B. subtilis* ATTC 6633 (MIC was 83.7 mg/mL), while the methanol extract did not show any antimicrobial potential [37]. Mihaylova et al. [37] used the same extraction techniques as in our research, pressurized liquid extraction, but with another solvent (ethanol–water). The antimicrobial activity against selected bacteria was observed by the inhibition zone [37], and therefore, no valid comparison can be made. According to Krivokapuć et al. (2020), the most effective antimicrobial activity was obtained by applying chloroform extract against Gram-positive bacteria, while there was no significant difference between water and methanol extracts regarding antimicrobial activity [26].

## 5. Conclusions

In order to isolate the molecules of interest to be used as food additives, it is necessary to apply adequate extraction techniques that preserve their antimicrobial potential. With specific focus on the utilization of “clean technologies” and by obtaining ready-to-use extracts, subcritical water extraction was shown to be an effective method for the isolation of bioactive molecules from *A. ursinum* with antimicrobial potential. Investigated opulent extract with polyphenolic and sulfur compounds showed to be effective against all examined foodborne bacteria. Accordingly, *A. ursinum* extract has the potential to be used as a natural antimicrobial additive. However, attention should be paid when incorporating the extract into food products. The level of natural additives of extract required for sufficient inhibition of microorganisms in foods may be considerably higher in comparison to laboratory media. *Allium ursinum* extract obtained with subcritical water extraction may be more suitable for implementation in food products using the extract obtained for example by maceration, as the subcritical one has several-fold higher content of bioactive compounds. In that sense, a lower amount of extract can be applied without altering food product characteristics. However, further research is needed to determine the optimum levels of *A. ursinum* subcritical water extract that can be safely applied in exact food systems and that can exhibit antimicrobial activity.

## Figures and Tables

**Table 1 microorganisms-10-01358-t001:** Chemical composition of *A. ursinum* extract detected by HPLC.

Detected Compounds	Concentration (μg/mL Extract)
Polyphenolic compounds	
Gallic acid	32.97 ± 5.21
Gallic acid derivate	9.10 ± 0.57
Gallic acid derivate	7.24 ± 1.09
Kaempferol derivate	8.96 ± 0.23
Kaempferol derivate	16.76 ± 1.04
Kaempferol derivate	9.48 ± 0.12
Kaempferol derivate	20.45 ± 2.56
Kaempferol derivate	29.95 ± 4.23
Catechin derivate	7.24 ± 0.89
Catechin derivate	6.89 ± 1.09
Catechin derivate	3.44 ± 0.89
Sulfur compounds	
S-methyl methanethiosulfonate	302.6 ± 10.12
Alilsulfid	44.1 ± 2.34
Diallyl disulfide	27.3 ± 2.12

Presented values of the observed parameters are written as the result of three measurements (*n* = 3) ± standard deviation.

**Table 2 microorganisms-10-01358-t002:** Antimicrobial activity of *A. ursinum* extract at concentrations of 5, 10 and 30 mg/mL.

Bacteria	LC (log_10_ cfu/mL)	*A. ursinum* Extract (mg/mL)			
		5	10	30	Positive Control
		Bacterial Count (log_10_ cfu/mL)			
*L. monocytogenes*	2	8.63 ^c^ ± 0.01	2.19 ^b^ ± 0.05	n.d. ^a^	8.64 ^c^ ± 0.02
	3	8.73 ^c^ ± 0.02	3.20 ^b^ ± 0.03	n.d. ^a^	8.74 ^c^ ± 0.01
	4	8.71 ^c^ ± 0.01	4.18 ^b^ ± 0.02	n.d. ^a^	8.71 ^c^ ± 0.01
	5	8.71 ^c^ ± 0.03	5.12 ^b^ ± 0.03	n.d. ^a^	8.71 ^c^ ± 0.02
	6	8.77 ^c^ ± 0.02	7.73 ^b^ ± 0.01	n.d. ^a^	8.75 ^c^ ± 0.02
*S*. Enteritidis	2	8.74 ^c^ ± 0.01	8.35 ^b^ ± 0.02	n.d. ^a^	8.78 ^c^ ± 0.03
	3	8.76 ^c^ ± 0.01	8.20 ^b^ ± 0.02	n.d. ^a^	8.76 ^c ±^ 0.01
	4	8.77 ^c^ ± 0.01	8.07 ^b^ ± 0.03	n.d. ^a^	8.78 ^c^ ± 0.02
	5	8.64 ^c^ ± 0.03	8.01 ^b^ ± 0.02	n.d. ^a^	8.64 ^c^ ± 0.04
	6	8.74 ^c^ ± 0.01	8.40 ^b^ ± 0.02	n.d. ^a^	8.74 ^c^ ± 0.02
*E. coli* 10536	2	8.91 ^c^ ± 0.03	8.39 ^b^ ± 0.02	n.d. ^a^	8.91 ^c^ ± 0.02
	3	8.90 ^c^ ± 0.02	8.36 ^b^ ± 0.01	n.d. ^a^	8.90 ^c^ ± 0.01
	4	8.91 ^c^ ± 0.03	8.40 ^b^ ± 0.04	n.d. ^a^	8.91 ^c^ ± 0.02
	5	8.74 ^c^ ± 0.03	8.01 ^b^ ± 0.02	n.d. ^a^	8.90 ^d^ ± 0.01
	6	8.64 ^c^ ± 0.04	8.40 ^b^ ± 0.02	n.d. ^a^	8.88 ^d^ ± 0.03
*E. coli* 8739	2	8.80 ^c^ ± 0.02	8.15 ^b^ ± 0.01	n.d. ^a^	8.80 ^c^ ± 0.01
	3	8.80 ^c^ ± 0.02	8.17 ^b^ ± 0.04	n.d. ^a^	8.80 ^c^ ± 0.03
	4	8.78 ^c^ ± 0.03	8.10 ^b^ ± 0.02	n.d. ^a^	8.79 ^c^ ± 0.04
	5	8.78 ^c^ ± 0.02	8.15 ^b^ ± 0.01	n.d. ^a^	8.79 ^c^ ± 0.03
	6	8.79 ^c^ ± 0.03	8.15 ^b^ ± 0.02	n.d. ^a^	8.88 ^d^ ± 0.02
*P. hauseri*	2	8.87 ^c^ ± 0.01	8.38 ^b^ ± 0.03	n.d. ^a^	8.87 ^c^ ± 0.02
	3	8.87 ^c^ ± 0.02	8.34 ^b^ ± 0.01	n.d. ^a^	8.87 ^c^ ± 0.01
	4	8.86 ^c^ ± 0.02	8.39 ^b^ ± 0.03	n.d. ^a^	8.87 ^c^ ± 0.03
	5	8.88 ^c^ ± 0.03	8.39 ^b^ ± 0.04	n.d. ^a^	8.88 ^c^ ± 0.01
	6	8.86 ^c^ ± 0.02	8.39 ^b^ ± 0.02	n.d. ^a^	8.87 ^c^ ± 0.01
*E. faecalis*	2	7.57 ^c^ ± 0.01	7.25 ^b^ ± 0.04	n.d. ^a^	7.58 ^c^ ± 0.03
	3	7.57 ^c^ ± 0.03	7.30 ^b^ ± 0.02	n.d. ^a^	7.57 ^c^ ± 0.04
	4	7.56 ^c^ ± 0.03	7.36 ^b^ ± 0.02	n.d. ^a^	7.57 ^c^ ± 0.02
	5	7.56 ^c^ ± 0.02	7.28 ^b^ ± 0.01	n.d. ^a^	7.56 ^c^ ± 0.03
	6	7.54 ^b^ ± 0.03	7.34 ^b^ ± 0.05	n.d. ^a^	7.54 ^c^ ± 0.01

Presented values of the observed parameters are written as a mean the result of three measurements (*n* = 3) ± standard deviation. Statistical significance is considered by the row. Different letters show statistically significant different means in rows of the observed data (*p* < 0.05), according to post hoc Tukey’s HSD test. LC, level of contamination; n.d., not detected.

**Table 3 microorganisms-10-01358-t003:** Antimicrobial activity of *A. ursinum* extract at concentrations of 26, 27, 28 and 29 mg/mL.

Bacteria	LC (log_10_ cfu/mL)	*A. ursinum* Extract Concentration (mg/mL)				
		26	27	28	29	Positive Control
		Bacterial Count (log_10_ cfu/mL)				
*L. monocytogenes*	2	1.27 ^b^ 0.02	n.d. ^a^	n.d. ^a^	n.d. ^a^	8.69 ^c^ ± 0.01
	3	2.70 ^b^ ± 0.04	n.d. ^a^	n.d. ^a^	n.d. ^a^	8.73 ^c^ ± 0.01
	4	3.90 ^b^ ± 0.02	n.d. ^a^	n.d. ^a^	n.d. ^a^	8.72 ^c^ ± 0.02
	5	5.00 ^a^ ± 0.01	4.93 ^a^ ± 0.01	1.20 ^c ±^ 0.05	n.d. ^b^	8.70 ^d^ ± 0.01
	6	7.52 ^d^ ± 0.02	6.92 ^c ±^ 0.02	2.78 ^b^ ± 0.03	n.d. ^a^	8.73 ^e^ ± 0.02
*S*. Enteritidis	2	7.95 ^d ±^ 0.03	3.70 ^c ±^ 0.04	1.52 ^b^ ± 0.06	n.d. ^a^	8.74 ^e^ ± 0.01
	3	8.00 ^d^ ± 0.01	4.55 ^c ±^ 0.02	3.31 ^b^ ± 0.04	n.d. ^a^	8.77 ^e^ ± 0.03
	4	8.15 ^d ±^ 0.02	5.87 ^c ±^ 0.01	4.10 ^b ±^ 0.03	n.d. ^a^	8.76 ^e^ ± 0.01
	5	8.24 ^d ±^ 0.01	7.00 ^c ±^ 0.02	5.40 ^b^ ± 0.04	n.d. ^a^	8.75 ^e^ (0.02)
	6	8.37 ^d ±^ 0.02	7.60 ^c ±^ 0.01	6.40 ^b ±^ 0.04	4.95 ^a^ ± 0.05	8.73 ^e^ ± 0.03
*E. coli* 10536	2	8.30 ^d ±^ 0.03	4.10 ^c ±^ 0.05	1.71 ^b^ ± 0.07	n.d. ^a^	8.92 ^e^ ± 0.01
	3	8.32 ^d ±^ 0.01	4.65 ^c ±^ 0.06	3.14 ^b^ ± 0.04	n.d. ^a^	8.90 ^e^ ± 0.03
	4	8.35 ^d ±^ 0.04	5.92 ^c^ ± 0.03	4.25 ^b^ ± 0.06	n.d. ^a^	8.92 ^e^ ± 0.02
	5	8.33 ^d^ ± 0.01	7.30 ^c^ ± 0.02	5.10 ^b^ ± 0.02	n.d. ^a^	8.91 ^e^ ± 0.01
	6	8.38 ^d^ ± 0.03	7.82 ^c^ ± 0.02	6.26 ^b^ ± 0.04	4.80 ^a ±^ 0.03	8.93 ^e^ ± 0.02
*E. coli* 8739	2	7.90 ^d ±^ 0.02	4.17 ^c^ ± 0.04	1.00 ^b^ ± 0.08	n.d. ^a^	8.81 ^e^ ± 0.01
	3	8.11 ^d^ ± 0.02	4.80 ^c^ ± 0.03	3.20 ^b^ ± 0.05	n.d. ^a^	8.80 ^e^ ± 0.03
	4	8.15 ^d^ ± 0.01	5.76 ^c^ ± 0.04	4.18 ^b^ ± 0.04	n.d. ^a^	8.82 ^e^ ± 0.02
	5	8.14 ^d^ ± 0.02	7.20 ^c^ ± 0.02	5.20 ^b^ ± 0.03	n.d. ^a^	8.79 ^e^ ± 0.01
	6	8.17 ^d^ ± 0.01	7.83 ^c^ ± 0.02	6.21 ^b^ ± 0.02	4.65 ^a^ ± 0.04	8.86 ^e^ ± 0.03
*P. hauseri*	2	8.29 ^d^ ± 0.02	4.22 ^c^ ± 0.03	1.79 ^b^ ± 0.05	n.d. ^a^	8.86 ^e^ ± 0.03
	3	8.32 ^d^ ± 0.01	4.74 ^c^ ± 0.03	3.56 ^b^ ± 0.02	n.d. ^a^	8.87 ^e ±^ 0.01
	4	8.31 ^d^ ± 0.01	5.81 ^c^ ± 0.01	4.87 ^b^ ± 0.03	n.d. ^a^	8.87 ^e^ ± 0.02
	5	8.35 ^d^ ± 0.01	7.15 ^c^ ± 0.02	5.33 ^b^ ± 0.04	n.d. ^a^	8.88 ^e^ ± 0.02
	6	8.38 ^d^ ± 0.03	7.80 ^c^ ± 0.01	6.44 ^b^ ± 0.02	4.72 ^a^ ± 0.03	8.88 ^e^ ± 0.02
*E. faecalis*	2	6.52 ^d^ ± 0.03	4.00 ^c^ ± 0.04	1.59 ^b^ ± 0.07	n.d. ^a^	7.56 ^e^ ± 0.03
	3	6.80 ^d^ ± 0.02	4.70 ^c^ ± 0.02	3.44 ^b^ ± 0.04	n.d. ^a^	7.58 ^e^ ± 0.01
	4	7.31 ^d^ ± 0.01	5.65 ^c^ ± 0.03	4.35 ^b^ ± 0.03	n.d. ^a^	7.57 ^e^ ± 0.02
	5	7.33 ^d^ ± 0.02	5.72 ^c^ ± 0.01	5.10 ^b^ ± 0.02	n.d. ^a^	7.58 ^e^ ± 0.01
	6	7.32 ^d^ ± 0.01	6.81 ^c^ ± 0.03	6.35 ^b^ ± 0.02	4.50 ^a^ ± 0.03	7.59 ^e^ ± 0.02

Presented value aremean values of the observed parameters are written as the result of three measurements (*n* = 3) ± standard deviation. Statistical significance is considered by the row. Different letters show statistically significant different means in rows of the observed data (*p* < 0.05), according to post hoc Tukey’s HSD test. LC, level of contamination; n.d., not detected.

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
