# Peer review of "Antibacterial Potential of Allium ursinum Extract Prepared by the Green Extraction Method"

_microorganisms, 2022, doi:10.3390/microorganisms10071358_

Round 1

Reviewer 1 Report

The Authors of the manuscript entitled “Antimicrobial potential of Allium ursinum extract prepared by green extraction method” present the antimicrobial activity of Allium ursinum extract obtained by subcritical water extraction method. Reported results contribute to the evaluation of the best extraction method in order to obtain a good antimicrobial activity together with a more sustainable approach to be used in food application.

The information provided would be useful to the scientific community but, in my opinion, is not enough to justify the publication of the manuscript in the present form in “Microrganisms”.

Some aspects have to be improved:

The Authors should well explain whether the compounds extracted by green method are more efficient to kill bacteria compared to those extracted by conventional methods. They write in the results section: “The content of total phenols was twice as high as the content of total phenols in the extract obtained by maceration, while the content of total flavonoids was about 3 times higher than maceration results” (this sentence needs to a reference about the total phenols and flavonoids reported by maceration for comparison). Is the higher phenols and flavonoids content responsible to a better antibacterial action with respect other methods? And, if extracted compounds are the same as those obtained by conventional methods, what is the novelty in their antibacterial action that has not been evaluated in previous works based on conventional extraction? Only the action on the contamination levels? Or is the antibacterial action comparable to the conventional methods but the proposed green method provides the same or better results (in terms of different compounds extracted and at higher concentrations and stability) with more safety for food application?  In addition, the Authors stated that allicin was not present in the extract composition. Sulfur compounds were at very low concentration. Since allicin and sulfur compounds play an important role in the antibacterial activity, is their loss a downside of the method?

The Authors should better address these aspects and provide critical comments on the advantages versus drawbacks of the subcritical water extraction procedure.

Specific Comments:

In the Materials and Methods section a paragraph should be introduced listing the Chemicals used for the experiments.

The extraction method indicates that the sample of Allium ursinum presents the same features of sample used by Tomsik et al. (2017). It’s the same? Is the procedure the same as in Tomsik et al., 2017 or has it been slightly modified?

It is not clear how the compounds and their concentrations were obtained. Data from HPLC analysis should be listed in a Table showing both compounds and metabolites. Respective concentrations together with the references used for identification should be added.

The positive controls of antibiotics in the antibacterial tests are missing. Authors have to include in the setting of antibacterial assays a set of experiments with antibiotics commonly used for bacteria tested, as positive control.

The antibacterial activity should be reported as MIC and MBC (mg/ml). Table 1 and Table 2 should report MIC and MBC values for the different bacteria and levels of contamination including extract concentrations omitted (15, 20, 25 mg/ml).

Finally, the reading of the manuscript is very difficult due to the numerous typing and grammar errors together with duplicated and incomprehensible sentences. A very careful revision of manuscript is needed.

Author Response

We want to thank the Reviewer for her/his thorough reading of our manuscript and all the useful comments and corrections which have been very helpful in improving the manuscript. We hope that the following answers will clarify the requested points.

Reviewer 2 Report

Manuscript "Antimicrobial potential of Allium ursinum extract prepared by green extraction method" presents interesting research results.

Detailed comments:

line 145 - American Type Culture Collection should be instead of American type culture collection.

Table 1 - why do the authors present the results of these studies only, if the methodology describes others? According to the CLSI and EUCAST standards, one concentration of microorganisms is used in antimicrobial tests, why do the authors present results for several concentrations?

Standard deviation results are usually given after the plus / minus sign, not in parentheses.

Authors should adapt the units in the manuscript to the journal requirements.

There is no table in the manuscript that would present the HPLC results.

Tables 1 and 2 present the same pattern of experiments and the authors may consider combining them into one.

The authors presented only the MIC and MBC studies, in subsequent manuscripts they may consider examining the mechanisms of the extract's action on microbes, e.g. by determining the time-kill, or checking the effect on the cell wall and membrane, as well as bacterial mobility.

The research is interesting and the fungus effects can be tested next time, meanwhile the title of the mancript can be changed to an antibacterial test, not an antimicrobial test.

Author Response

(The authors gave the same response as above.)

Reviewer 3 Report

The manuscript described the antimicrobial activity of  Allium ursinum extract. It is interesting work, but I have a few comments:

Line 18: L. monocytogenes, should be a full name, Listeria monocytogenes

Line 20: “L. monocytogenes” should be italic;

Line 20:  „mg mL-1” should be a dot between mg and mL-1 in whole manuscript

Line 22:  “29 and 30 mg mL-1. respectively.”, should be a comma after mL-1 instead of dot

Line 92: “coleted” should be collected ?

Line 102: “as as solvent” – double “as”

Line 111: “performed” should be ‘performed’

Line 113: “Prior analysis A. ursinum extracts of…”, should be ‘Prior analysis of A. ursinum extracts…’

In manuscript  there are many typo’s, e.g.:

Line 195: typo in word “bilogical”

Line 198: typo in word “comaparing”

Line 199: typo in word “contet”

Line 211; typo in word “catehins”

Line 215: “subrictivcal water” - typo

Line 282: typo in word „patgogens”

Furthermore:

Line 342: “A. ursinum” should be italic

Line 343 and also in line 377: a typo in the name A. ursinum/A.ursinm

Author Response

(The authors gave the same response as above.)

Round 2

Reviewer 1 Report

The authors have introduced most of the required changes to the manuscript. However, I think the manuscript needs an extensive spell check before publication in Microrganisms

Reviewer 2 Report

It seems that most of the corrections were introduced by the authors, but the changes were not marked, which makes it very difficult for the reviewer to analyze the changes in detail.